# Constitutively Activating Mutants of Equine LH/CGR Constitutively Induce Signal Transduction and Inactivating Mutations Impair Biological Activity and Cell-Surface Receptor Loss In Vitro

**DOI:** 10.3390/ijms221910723

**Published:** 2021-10-03

**Authors:** Munkhzaya Byambaragchaa, Hoon-Ki Seong, Seung-Hee Choi, Dae-Jung Kim, Myung-Hwa Kang, Kwan-Sik Min

**Affiliations:** 1Institute of Genetic Engineering, Hankyong National University, Ansung 17579, Korea; monkhzaya_b@gmail.com (M.B.); whimoon17@gmail.com (H.-K.S.); 2Animal Biotechnology, Graduate School of Future Convergence Technology, Hankyong National University, Ansung 17579, Korea; seunghee2105@gmail.com; 3Jeju Fisheries Research Institute, National Institute of Fisheries Science (NIFS), Jeju 63610, Korea; djkim4128@korea.kr; 4Department of Food Science and Nutrition, Hoseo University, Asan 31499, Korea; mhkang@hoseo.edu

**Keywords:** eLH/CGR, constitutively activating mutation, inactivating mutation, cAMP response, cell-surface loss of receptor

## Abstract

The signal transduction of the equine lutropin/choriogonadotropin receptor (eLH/CGR) is unclear in naturally occurring activating/inactivating mutants of this receptor, which plays an important role in reproductive physiology. We undertook the present study to determine whether conserved structurally related mutations in eLH/CGR exhibit similar mechanisms of signal transduction. We constructed four constitutively activating mutants (M398T, L457R, D564G, and D578Y) and three inactivating mutants (D405N, R464H, and Y546F); measured cyclic adenosine monophosphate (cAMP) accumulation via homogeneous time-resolved fluorescence assays in Chinese hamster ovary cells; and investigated cell-surface receptor loss using an enzyme-linked immunosorbent assay in human embryonic kidney 293 cells. The eLH/CGR-L457R-, -D564G-, and -D578Y-expressing cells exhibited 16.9-, 16.4-, and 11.2-fold increases in basal cAMP response, respectively. The eLH/CGR-D405N- and R464H-expressing cells presented a completely impaired signal transduction, whereas the Y546F-expressing cells exhibited a small increase in cAMP response. The cell-surface receptor loss was 1.4- to 2.4-fold greater in the activating-mutant-expressing cells than in wild-type eLH/CGR-expressing cells, but was completely impaired in the D405N- and Y546F-expressing cells, despite treatment with a high concentration of agonist. In summary, the state of activation of eLH/CGR influenced agonist-induced cell-surface receptor loss, which was directly related to the signal transduction of constitutively activating mutants.

## 1. Introduction

The receptor of the pituitary and placental gonadotropin in species expressing placental chorionic gonadotropin (CG) during early pregnancy plays a critical role in reproductive physiology. Equine CG (eCG), a unique member of the gonadotropin family, displays both luteinizing hormone (LH)-like and follicle-stimulating hormone (FSH)-like activities in non-equid species [1,2]. However, eCG, secreted from endometrial cups during early pregnancy, exhibits only LH-like activity in equine species [3]. The β-subunits of eCG and eLH are encoded as one gene, whereas their expression differs in the placenta and pituitary gland. Additionally, rec-eCG exhibits dual activities of LH and FSH in other species [1,3]. Thus, both of them bind to the same receptor: the lutropin/choriogonadotropin receptor (LH/CGR). LHR and FSHR, termed gonadotropin receptors, belong to a family of seven transmembrane G protein-coupled receptors (GPCR) whose primary function is the mediation of the signal pathway by LH, CG, and FSH in the gonads [4]. The genes that encode GPCRs form one of the largest gene families [5]. eLH/CGR belongs to a subgroup of glycoprotein hormone receptors within the GPCR family, and possesses a large extracellular domain [6,7,8].

The gonadotropin receptors of mammals (human, rat, and equine) and fish (eel) are highly homologous, with the highest level of amino acid conservation within the transmembrane helices [4,9,10]. The agonist–receptor complex is internalized by a dynamin-dependent pathway and moved to the endosome compartment without agonist dissociation [11]. The receptor-mediated signaling displays two important processes: cAMP production by adenylyl cyclase and agonist-induced down-regulation of the cell-surface receptors for regulating cellular responsiveness [12]. In the past several years, point mutations of human LHR (hLHR) and rat LHR (rLHR) have been reported to exhibit constitutive activation and impair signal transduction [11,13]. Specifically, the LHR gene has been associated with an abundance of naturally occurring mutations related to reproductive failure in mammals [8,14].

In humans, hLHR-M398T, hLHR-L457R, hLHR-D564G, and hLHR-D578Y or hLHR-D578G mutants, known as naturally occurring mutations, have been reported to display constitutive activation, leading to an increase in the basal cAMP response without agonist treatment [14,15,16]. In rats, rLHR-L435R (equivalent to L457R in eLH/CGR) and rLHR-D556Y or rLHR-D556G mutants (equivalent to D578Y in eLH/CGR) induced constitutive activation of rLHRs, showing a considerable increase in cAMP levels in cells incubated without hormones [11]. The inactivating mutants rLHR-D383N (equivalent to D405N in eLH/CGR), rLHR-R442H (equivalent to R464H in eLH/CGR), and Y524F mutants (equivalent to Y546F in eLH/CGR) have been reported to impair signal transduction under treatment with high concentrations of agonist [11,17], decreasing the rate of ligand internalization, while activating mutants enhanced the rate of hormone internalization.

In recent years, GPCR signal transduction has been studied in detail with respect to cell-surface receptor loss, constitutive internalization, constitutive endocytosis, recycling, and β−arrestin-dependent internalization [18,19,20,21,22]. Many studies have elucidated several features of the post-endocytotic trafficking of LHR and FSHR [23,24]. Recently, we identified several characteristics of signal transduction of the recombinant eCG through eLH/CGR—the function of constitutively activating eFSHR mutants, eel LHR and eel FSHR—demonstrating that activating mutants produces a considerable increase in the basal cAMP response [9,10,25,26]. Although various agonists have been reported to induce the signal transduction of glycoprotein hormone [17,27,28], and several studies have been reported on dual biological activities and glycosylation site roles for rec-eCG, there are few reports on the characterization and function of cellular signal transduction of eLH/CGR in naturally occurring activating/inactivating mutants. We designed the present study to investigate the possibility that the activating signal transduction of eLH/CGRs is necessary for the loss of cell-surface receptors in the presence of mutations in amino acid residues that are highly conserved among GPCRs and that have been previously reported to activate/inactivate GPCRs. We hypothesized that mutations of eLH/CGR that render it constitutively active enhance the cell-surface loss of the receptor.

In the present study, we aimed to delineate the mechanism of cell-surface receptor loss by evaluating the effect of single point mutations of seven distinct amino acid residues that are highly conserved among glycoprotein hormone receptors including LHR, FSHR, and TSHR. These mutations, which stimulate basal cAMP responsiveness and/or attenuate agonist-induced activation of the receptor, were four constitutively activating eLH/CGR mutations (M398T, L457R, D564G, and D578Y) and three inactivating mutations (D405N, R464H, and Y546F). Our study revealed a marked constitutive basal cAMP response and rapid cell-surface receptor loss in cells expressing the activating eLH/CGR mutants.

## 2. Results

### 2.1. Preparation and Cell-Surface Expression of Wild-Type eLH/CGR and the Mutant Receptors

To determine how eLH/CGR affects hormone-receptor interaction, we generated four constitutively activating mutations of eLH/CGR in the II, III, and VI transmembrane helices of eLH/CGR (M398T, L457R, and D578Y) and intracellular domain three (D560G). We additionally constructed three inactivating mutations: D405N and Y546F in the II and V transmembrane helices, and R464H in the extracellular domain II (Figure 1).

The surface expression of eLH/CGR mutants was determined via an enzyme-linked immunosorbent assay (ELISA) in transiently transfected Chinese hamster ovary (CHO)-K1 cells (Figure 2). The expression level of wild-type eLH/CGR was considered to be 100%, and that of L457R, D564G, and D578Y were approximately 31.2%, 60.6%, and 47.9%, respectively. In contrast, the expression level of the M398T mutant was the lowest, at 4.7%, relative to that of the wild-type receptor. In the inactivating mutants, the expression levels for D405N and Y546F were 80.2% and 90.1%, respectively. However, the expression of the R464H mutant was nearly 3.8%, similar to that of M398T, which was an activating mutant. However, several differences were unexpectedly detected on the cell-surface expression. We subsequently determined the cAMP response and cell-surface receptor loss induced by agonist treatment.

### 2.2. cAMP Responsiveness Induced by Agonist in Activating Mutants and Inactivating Mutants

The effects of activating mutations on the basal and eCG-stimulated cAMP responsiveness are summarized in Figure 3 and Table 1. The basal and Rmax cAMP responses in the wild-type receptor were 1.5 and 85.3 nM/10^4^ cells, respectively. cAMP production increased in a dose-dependent manner. The half maximal effective concentration (EC_50_) value of the eCG-stimulated cAMP response was approximately 44.2 ng/mL. The activating mutants (L457R, D564G, and D578Y) induced constitutive activation of eLH/CGR, as illustrated by 16.9-, 16.4-, and 11.2-fold increases in the basal cAMP response without agonist treatment; however, the M398T mutant was not induced.

This elevated basal response of cAMP in the L457R mutant corresponds to approximately 30% of the maximal response detected in wild-type eLH/CGR cells; however, it was 63% of the maximal response observed with the L457R mutants. eLH/CGR-L457R do not show any increase in eCG-stimulated cAMP responses over that already produced as a result of constitutive activity (Figure 3). The L457R mutant was specifically unresponsive to treatment with a high concentration of agonist. However, the other mutants (M398T, D564G, and D578Y) cause an increase in eCG-stimulated cAMP production above those levels synthesized as a result of constitutive activity. The cAMP production for the M398T mutant upon treatment with a high concentration of agonist was slightly below that in cells expressing wild-type eLH/CGR. The D564G and D578Y mutants induced constitutive activation of the eLH/CGR, as illustrated by a considerable increase in basal cAMP level in cells incubated without the eCG agonist. This elevated basal cAMP production corresponds to only 17–27 % of the maximal response. Nevertheless, both mutants responded to eCG with a robust increase in cAMP accumulation, which correlated with the result for wild-type eLH/CGR.

The eCG/LHRs with the inactivating mutations—D405N, R464H, and Y546F—were evaluated by quantifying cAMP accumulation in cells incubated with increasing concentrations of eCG. The basal cAMP response was not affected by the inactivating mutation (Figure 4 and Table 2). As predicted, signaling was completely impaired in these mutant receptors (D405N, R464H, and Y546F). The Y546F mutant showed a small increase under stimulation with a high concentration of eCG; however, the maximal response to eCG was only 26% of the corresponding values in the wild-type eCG/CGR (Table 2). The basal and maximal cAMP responses with eCG agonist treatment were subjected to comparative analysis (Figure 5). We observed a specific increase in basal cAMP production by the cells expressing L457R and D564G mutants (Figure 5). However, the L457R mutant displayed the lowest maximal response (47%) among the activating mutants compared to wild-type eLH/CGR. In contrast, the maximal cAMP response for the D564G and D578Y mutants was approximately 106% and 117%, respectively, of that of wild-type eLH/CGR (Figure 5). In agreement with our previous studies on the equivalent mutation of rLHR and eel LHR, the inactivating mutants completely impaired the maximal cAMP response; they presented 4–26 % of the maximal response of the wild-type receptor (Figure 5). 

The difference in basal cAMP response and EC_50_ values for the wild-type receptors in Table 1 and Table 2 was attributed to the fact that the experiments were performed separately. 

### 2.3. Cell-Surface Receptor Loss Induced by Treatment with the eCG Agonist

Next, to accurately analyze the rate of the cell-surface loss of receptors, we performed experiments in which the cell-surface loss of the receptor was measured in a time-dependent manner in the continuous presence of eCG. The results for the four activating and three inactivating mutants are shown in Figure 6 and Figure 7.

Cell-surface expression in the wild-type eLH/CGR cells gradually decreased until it reached approximately 69% of the pre-treatment value at 15 min. Subsequently, the value was consistent until 240 min to the end point. Cells expressing the eLH/CGR-M398T and eLH/CG-L457R mutants exhibit faster rates than cells expressing wild-type eLH/CGR, demonstrating that the cell-surface loss of receptors reached 50% and 64%, respectively, at 15 min. In cells harboring the D564G activating mutation, the loss of cell-surface receptors rapidly decreased to 49% in the first 15 min, and subsequently remained between 32% and 38% for 240 min. The D564G mutant exhibited the most rapid decrease in cell-surface receptor loss among the activating mutants. Additionally, the cell-surface receptor loss was considerably similar between cells expressing the D578Y mutant (52%) and those expressing D564G. The cell-surface loss of the receptor for all activating mutants was faster than for wild-type eLH/CGR. The surface loss of the receptor was not observed in the cells expressing the inactivating mutants (D405N and Y546F) and was considerably slower in the cells expressing the R464H mutant than in those expressing the wild-type receptor (Figure 7). 

Loss of the expression of cell-surface receptors at 30 min (Figure 8) decreased considerably in the wild-type eLH/CGR (39%) compared with that in control cells pre-incubated in the absence of the eCG agonist (considered as 0% of loss of surface receptor). In cells harboring the M398T and L457R mutants, the loss was slightly increased (47% and 43%) compared to that observed for the wild-type receptor. Additionally, the D564G and D578Y mutants presented a higher increase in the cell-surface loss (62% and 55%, respectively) than wild-type eLH/CGR. 

The rate of formation of the agonist–receptor complexes induced by the constitutively activating and inactivating mutants of eLH/CGR described above are presented in Table 3 and Table 4. The rates of loss of cell-surface agonist–receptor complexes in both the wild-type and activating-mutant were very rapid (3.2~7.6 min; Table 3). Specifically, the L457R mutant exhibited the most rapid rates (3.2 min) of cell-surface loss of the receptor among the activating mutants, despite the absence of a further increase in cAMP response induced by treatment with a high concentration of eCG agonist. These data clearly show that the three inactivating mutations—eLH/CGR-D405N, eLH/CGR-R464H, and eLH/CGR-Y546F—significantly reduced the rate of cell-surface loss of the receptor, whereas the activating mutations evaluated in the present study enhanced the rate of cell-surface loss of eLH/CGR. Thus, the loss of cell-surface receptors is consistent with the cAMP responsiveness induced by treatment with the eCG agonist.

For cells expressing wild-type receptors, the values of t*_1/2_* and maximum reduction presented in Table 3 and Table 4 were different because the experiments were conducted independently.

## 3. Discussion

The present study showed that the four mutations—eLH/CGR-M398T, L457R, D564G, and D578Y—resulted in a distinctly increased cAMP response without agonist treatment, suggesting that these mutations might produce constitutively activating mutants of eLH/CGRs. Our previous observations have suggested that the same active conformations of rLHR [11], eel FSHR [10], and eFSHR [25] are involved in the stimulation of G proteins and loss of the cell-surface receptor in ligand–receptor complexes. 

In the present study, the M398T mutant exhibited a 1.4-fold increase in basal cAMP response, consistent with the suggestion that the hLHR mutant exhibited a 3.5-fold increase in basal cAMP response in COS-7 cells [14] and a similar increase in HEK 293 cells [4]. However, another study demonstrated a dramatic increase in basal cAMP response of 25-fold in HEK 293 cells [7]. Thus, we suggest that the eLH/CGR-M398T mutant displays a constitutive activation of cAMP response without agonist treatment, and the basal cAMP response differs from that in the cells expressing those mutants, despite the small increase observed in this study. In agreement with previous studies on the equivalent mutation of the hLHR and rLHR [11,29,30], our results showed that the eLH/CGR-D540G and eLH/CGR-D578Y mutations induce a marked increase in cAMP production without agonist treatment. Compared to the wild-type eLH/CGR, the two mutants resulted in a 16.4- and 11.2-fold increase in basal cAMP production in CHO-K1 cells, indicating that such mutants are constitutively activating, as previously reported for other mammalian hLHR [13,30], and rLHR [11,31]. The maximal responses of these two mutants exhibit a 6–17% increase, relative to the cAMP level detected in cells expressing the wild-type eLH/CGR. A previous study on transgenic rat LHR-D556H (equivalent to D578Y in eLH/CGR) under control of the inhibin α−subunit promoter demonstrated that the level of constitutive activity was similar to the results obtained in vitro [6]. 

In the activation model, the eLH/CGR-L457R mutant exhibited the highest increase in basal cAMP response. However, eLH/CGR-L457R did not further increase cAMP accumulation. These results are consistent with data from the previous studies, indicating that hLHR-L457R [4,12,15,32,33], rLHR-L435R [11], and hFSHR-L469R [34] are constitutively activating with respect to the basal cAMP response without agonist treatment. As shown in this study, the basal cAMP response of the L457R mutant dramatically increased, whereas the maximal cAMP response corresponded to approximately 47% of that detected in cells expressing wild-type eLH/CGR (Figure 3 and Table 1). This lack of hormonal responsiveness has not been reported because of an impairment in maintaining the same hormonal binding affinity as the wild-type hLHR [35]. The L457R mutation is an unusual activating mutant of LHR, displaying particularly strong constitutive activity, but no further stimulation of Gs with high concentrations of ligand. Thus, we suggest that the eLH/CGR-L457R mutant is a significant model for determining the cellular mechanisms of eLH/CGR activation with/without high agonist treatment.

As predicted from the results described above, the mutations investigated in the present study (eLH/CGR-D405N, eLH/CGR-R464H, and eLH/CGR-Y546F) completely impair signal transduction. Thus, our results are consistent with previously reported signal-transduction studies on inactivating mutants in hLHR [17] and rLH [11]. Two inactivating mutants (D405N and Y546F) exhibit full cell-surface expression, but not loss of the cell-surface receptor by further agonist treatment. Conformational changes in the mutated receptors could explain why the inactivating mutants did not produce cAMP responses and impair the loss of the cell-surface receptor despite prolonged agonist stimulation.

According to the present results, the rates of cell-surface loss of the receptor in all activating mutants (3.2~5.3 min) were faster than that observed in cells expressing the wild-type receptor (7.6 min), clearly indicating important correlations between basal cAMP response and cell-surface receptor loss, except for the M398T mutant. The *t*_1/2_ value in two mutants (M398T and L457R) indicated faster cell-surface receptor loss than that in the other mutants (D564G and D578Y). Specifically, the M398T mutant did not exhibit any increase in the basal cAMP response; however, the *t_1/2_* value for this mutant indicated a 2.2-fold faster loss of the cell-surface receptor compared to those of the wild-type receptor. Thus, these differences between basal cAMP response and the cell-surface loss of receptors need to be clearly demonstrated.

However, the loss of the cell-surface receptor for the D564G and D578Y mutants at 30 min was higher than that for M398T and L457R. Thus, we suggest that, when induced by agonist treatment, the two groups do not present a conformation equivalent to that of eLH/CGRs. Our results were consistent with previous data reported by our colleagues and our equivalent studies on rLHR, eel LHR, and hLHR, which demonstrated that four signal-activating mutants—eLH/CGR-M398T, eLH/CGR-L457R, eLH/CGR-D564G, and eLH/CGR-D578Y—enhanced the cell-surface loss of the receptor [10] and increased the rate of internalization of the bound hCG [11]. We suggest that the loss of the cell-surface receptor is necessary for signal transduction, suggesting that inactivating mutants completely impair cAMP response and the loss of the cell-surface receptor. 

Many GPCRs are internalized into endosomes via a clathrin-dependent pathway and partly degraded in lysosomes or recycled to the cell membrane for prolonged agonist stimulation [36,37,38]. In the present study, the *t*_1/2_ rate indicated slightly faster cell-surface receptor loss in cells expressing the L457R mutant than in cells expressing the wild-type receptor, whereas the rate of loss in the former was slower than for wild-type eLH/CGR, D564G, and D578Y after 30 min. A possible explanation for these results is that both wild-type hLHR and the hLHR-L457R mutant are trafficked through different endosomal compartments, indicating that the hCG/hLHR-wt complex could follow a fast-recycling pathway from the early endosomes, whereas the hCG/hLHR-L457R could follow a slower recycling pathway involving either the recycling and/or the late endosomes. Thus, the L457R mutant is not routed to the lysosomes; most of it is recycled to the cell surface, and hormonal degradation is barely detectable [12]. The study of both hLHR-L457R and hFSHR-L460R have demonstrated that the increase in basal activities is highly similar between the two, and is observed for the strongest constitutively activating mutants; any differences may be attributed to differences in the shape and electrostatic features of the solvent-exposed cytosolic receptor domains involved in the receptor-G protein interface [4]. Our results indicated that the rates of internalization in the activating receptors (rLHR-L435R and rLHR-D556Y) were 17- and 2.6-fold faster than that observed in the agonist-occupied wild-type rLHR [11], and the inactivating mutants (rLHR-D383N, rLHR-R442H, and rLHR-Y524F) exhibited slower internalization, the half-life of the internalization process, exhibiting approximately 4-, 2-, and 1.9-fold increases following treatment with the agonist [11,17].

Therefore, we suggest that the rate of internalization and the cell-surface loss of the receptor of the constitutively activating mutant are strongly correlated, similar to that for the agonist-occupied wild-type eLH/CGR, whereas the inactivating mutants—D405N, R464H, and Y546F—completely impaired the cAMP response and the loss of the cell-surface receptor. Although the mechanism of the L457R mutant is not well understood, we suggest that the hormone-independent activity of eLH/CGR-L457R may include the formation of structural links between the transmembrane helices [15]. The cell-surface loss of the receptor, internalization, and the trafficking of new receptors to the cell membrane could not significantly affect the level of cell-surface receptor expression (Figure 2). β-arrestin, ubiquitously expressed intracellular regulators of the GPCR trafficking, pathway plays a key role in the down-regulation of the GPCR and its internalization. G protein and β-arrestin signaling mediate distinct physiological effects [38,39]. However, β-arrestin activation by eLH/CGR has not been well studied. Research is currently underway to clarify our theory on GPCR internalization, degradation, and recycling. Recently, biased allosteric modulators and regulators of other GPCR have been reported [36,37,40]. In the present study, the mutants located in the transmembrane domain, and between intracellular loops and the transmembrane domain, demonstrated that eLH/CGRs may be involved in interactions with different types of G-proteins. However, further molecular studies will be required for elucidating the functional interactions of the eLH/CGR–eCG complex in cells expressing constitutive activating and inactivating mutants.

## 4. Materials and Methods

### 4.1. Materials

The cloning vector (pGEM-T easy) was purchased from Promega (Madison, WI, USA). The pcDNA3 expression vector, FreeStyle CHO-suspension (CHO-S) cells, FreeStyle™ MAX reagent, and Lipofectamine-3000 were provided by Invitrogen (Carlsbad, CA, USA). The pCORON1000 SP VSV-G tag expression vector was purchased from Amersham Biosciences (Piscataway, NU, USA). OptiMEM medium, Ham’s F-12 medium, and serum-free CHO-S-SFM II were purchased from Gibco BRL (Grand Island, NY, USA). Restriction enzymes, polymerase chain reaction (PCR) reagents, and DNA ligation kit were purchased from Takara (Shiga, Japan). The primary rabbit anti-VSVG antibody and secondary HRP-conjugated anti-rabbit antibody were obtained from Abcam (Burlingame, CA, USA). SuperSignal enzyme-linked immunosorbent assay (ELISA) Femto Maximum substrate was obtained from Thermo Fisher Scientific (Waltham, MA, USA). CHO-K1 and HEK 293 cells were obtained from the Korean Cell Line Bank (KCLB, Seoul, Korea). The cAMP Dynamic 2 competitive immunoassay kit was purchased from Cisbio (Codolet, France). The QIAprep-spin plasmid kit was purchased from Qiagen Inc. (Hilden, Germany). The glass spinner flasks and disposable flasks were provided by Corning Inc. (Corning, NY, USA). The PMSG ELISA kit was purchased from DRG International Inc. (Mountainside, NJ, USA). All other reagents used were purchased from Sigma-Aldrich (St. Louis, MO, USA) and Wako Pure Chemicals (Osaka, Japan).

### 4.2. Site-Directed Mutagenesis and Vector Construction

To construct point mutations, we introduced the cDNA, encoding the full-length eLH/CGR using an overlap extension PCR strategy to create activating and inactivating mutants, as previously described [10]. Two different sets of PCRs were performed, and the primer sequences used in these experiments are shown in Table 5. The full-length PCR products were cloned into a pGEM-T easy vector, and the sequence of the entire region of each mutant generated by PCR was confirmed by DNA sequencing. We comparatively analyzed the conserved sequencing regions of the glycoprotein hormone receptors in mammalian, and selected the naturally occurring mutation sites in the transmembrane regions (II, III, V, and VI) and intracellular regions (II and III). Figure 1 presents a schematic representation of the naturally occurring mutation sites for four activating mutations (M398T, L457R, D564G, and D578Y) and three inactivating mutations (D405N, R464H, and Y546F) of eLH/CGR in the present study.

### 4.3. Production of Recombinant-eCGβ/α Mutants in CHO Suspension Cell

For rec-eCGβ/α production, the expression vectors were transfected into CHO-S cells using the FreeStyle™ MAX reagent, following the manufacturer’s instructions [41,42]. Briefly, the CHO-S cells were cultured in FreeStyle CHO expression medium at 1 × 10^7^ cells per 30 mL of medium for 3 days. On the day before transfection, the cells were passaged at a density of 5–6 × 10^5^ cells/mL with the medium (125 mL) in a disposable spinner flask.

A plasmid DNA (260 μg) and FreeStyle™ MAX reagent (260 μL) was diluted with Opti-PRO™ SFM to produce a total volume of 8 mL, which was gently mixed by inverting the tube. The mixed DNA-FreeStyle™ MAX was incubated for 10 min at room temperature for allowing complex formation, and slowly added to the suspension flask. The cell cultures were incubated at 37 °C in a humidified atmosphere comprising of 8% CO_2_ on an orbital shaking platform rotating at 135 rpm. Finally, the culture medium was collected on day 7 post-transfection and centrifuged at 100,000 × *g* for 10 min at 4 °C for removal of cell debris. The supernatant was collected and stored at −80 °C until analysis. The samples were concentrated using either a Centricon filter or by freeze-drying and mixing with phosphate-buffered saline (PBS). The sample was mixed 10–20 times, and the concentration of rec-eCGβ/α was determined via ELISA [1,2].

The rec-eCGβ/α protein was quantified using PMSG with anti-PMSG monoclonal antibody and horseradish peroxidase (HRP)-conjugated antibody, according to the manufacturer’s instructions. The culture medium (100 μL) was dispensed into the wells of 96-well plates coated with a monoclonal antibody. The samples were incubated for 60 min at room temperature, followed by incubation with 100 μL of HRP-conjugated secondary antibody. The samples were washed and incubated with the tetramethylbenzidine (TMB) substrate solution (100 μL) for 30 min at room temperature. The reaction was stopped by adding 50 μL of 1 M H_2_SO_4_. Absorbance at 450 nm was measured within 30 min using a microplate reader (Cytation 3; Biotek, Winooski, VT, USA). Finally, 1 IU was assumed to be 100 ng, on the basis of the conversion factor for the suggested assay protocol.

### 4.4. Transient Transfection into CHO-K1 Cells and HEK 293 Cells

The CHO cells were transfected using liposomal transfection [26]. The CHO cells were cultured in growth medium (Ham’s F-12 media supplemented with penicillin (50 U/mL), streptomycin (50 μg/mL), glutamine (2 mM), and 10% fetal bovine serum). The HEK 293 cells were cultured in growth medium (Dulbecco’s modified Eagle’s medium containing 10 mM Hepes, 50 μg/mL gentamycin, and 10% fetal bovine serum).

The CHO cells and HEK 293 cells were grown to 80–90% confluence in 6-well plates, and the plasmid DNAs were transfected using the Lipofectamine reagent. After the diluted DNA had been combined with Lipofectamine samples, the mixture was incubated for 20 min. The cells were washed with Opti-MEM, and the DNA-Lipofectamine complex was added to each well. After 5 h, growth medium containing 20% fetal bovine serum was added to each well. The CHO cells were used for cAMP analysis 48 h post-transfection. The HEK 293 cells were used for investigating the cell-surface loss of the receptor.

### 4.5. cAMP Analysis by Homogeneous Time-Resolved Fluorescence (HTRF) Assays

cAMP accumulation in CHO-K1 cells expressing wild-type eLH/CGR or the eLH/CGR mutants was measured using cAMP Dynamic 2 competitive immunoassay kits [10]. The transfected cells were seeded in a 384-well plate (10,000 cells per well). The standard samples were prepared to cover an average cAMP concentration of 0.17–712 nM (final concentration of cAMP per well). We added MIX to the cell dilution buffer to prevent cAMP degradation. To each well, 5 μL of compound medium buffer containing rec-eCG mutants was added. The plate was sealed and incubated for cell stimulation at room temperature for 30 min. The samples were incubated with the detection reagents, cAMP-d2 and anti-cAMP-cryptate (diluted five-fold in lysis buffer, 5 μL/well), for 1 h at RT. cAMP was detected by measuring the decrease in homogeneous time-resolved fluorescence (HTRF) energy transfer (665 nm/620 nm) using an Artemis K-101 HTRF microplate reader (Kyoritsu Radio, Tokyo, Japan). The specific signal-Delta F (energy transfer) is inversely proportional to the concentration of cAMP in the standard or the sample. The results were calculated on the basis of the 665 nm/620 nm ratio, and expressed as Delta F% (cAMP inhibition), according to the following equation: (Delta F% = (standard or sample ratio-mock transfection) × 100/mock transfection). The cAMP concentrations for the Delta F% values were calculated in nM using the GraphPad Prism software (GraphPad, Inc., La Jolla, CA, USA).

### 4.6. Agonist-Induced Cell-Surface Loss of Receptor

Loss of eLH/CGR from the cell-surface was assessed using ELISA [10,43]. HEK 293 cells were transfected with each mutant plasmid; subsequently, the cells were split into 96-well dishes (1 × 10^4^ cells) and coated with poly-d-lysine 24 h post-transfection. In the experiment to determine cell-surface loss, the cells were pre-incubated with or without rec-eCG (1000 ng/mL) for 30 min at 37 °C. The cells were incubated with 1000 ng/mL rec-eCG for the time-dependent tests (5, 15, 30, 60, 120, and 240 min). 

Briefly, the cells were fixed using 4% paraformaldehyde in Dulbecco’s PBS (DPBS) for 5 min at 25 °C. After three washes with DPBS, the wells were incubated with blocking solution (Tris-buffered saline with 1% bovine serum albumin) for 30 min. The primary antibody reaction was performed using a rabbit anti-VSVG antibody (1:1000), followed by incubation with an HRP-conjugated anti-rabbit antibody (1:15,000). The wells were washed four times with blocking solution. To each well, 80 μL of DPBS and 10 μL of SuperSignal ELISA Femto Maximum substrate were added for detection. Luminescence was measured using a Cytation 3-plate reader (BioTek, Winooski, VT, USA). The expression levels of the wild-type receptors were set as 100%. The cell-surface loss of wild-type and mutant eLH/CGRs was calculated by comparing the levels in the presence of rec-eCG to the levels in the non-treatment cells (taken as 0% of the loss of cell-surface receptors). 

### 4.7. Data Analysis

The Multalin multiple sequence alignment software was used for sequence analysis. GraphPad Prism 6.0 (San Diego, CA, USA) was used for analyzing cAMP responsiveness, EC_50_ values, and the stimulation curve analyses. Curves fitted in a single experiment were normalized to the background signal measured for the mock-transfected cells (Figure 3 and Figure 4). The values for cAMP levels and cell-surface receptors in the mock-transfected cells were subtracted from the corresponding values in the transfected cells. One-way ANOVA and Tukey’s multiple comparison tests were used for comparing the results between samples, using the GraphPad Prism 6.0 software. A *p*-value of <0.05 was considered to indicate a significant difference between groups.

## 5. Conclusions

This study shows that constitutively activating mutations of eLH/CGR (M398T, L457R, D564G, and D578Y) resulted in a significant increase in the basal cAMP production and a faster loss of the cell-surface receptor compared to wild-type eLH/CGR despite low cell-surface expression, which has been reported for mutations of these highly conserved amino acids in mammalian LHRs. The rate of loss of the M398T and L457R mutants from the cell surface was found to be very similar to that of wild-type eLH/CGR, whereas the half-life (*t*_1/2_) of the cell-surface receptor loss indicated a faster loss for the former than for wild-type eLH/CGR. The other two activating mutants (D564G and D578Y) exhibited a slightly faster rate of cell-surface receptor loss. However, the loss of the cell-surface receptor considerably decreased to 55–62% until termination after 15 min. Thus, we suggest that D564G and D578Y mutants exhibit a similar cAMP response as wild-type eLH/CGR for the constitutive loss of the cell-surface receptor. In contrast, the inactivating mutations (D405N, R464H, and Y546F) completely impaired the signal transduction of the agonist-mediated receptor response. The loss of the D405N and Y546F mutants from the cell-surface receptor was not complete; however, the loss of the R464H mutant was considerably slower than that of the agonist-occupied wild-type receptor. Thus, we suggest that the activation process might involve an agonist-induced conformational change in the eLH/CG receptor-eCG complex. These findings are extremely important for our understanding of eLH/CGR function and regulation with respect to mutations of highly conserved amino acids in mammalian glycoprotein hormone receptors. Future studies on the mutations of glycoprotein hormone receptors could provide highly useful information for identifying the cellular mechanism responsible for the structure–function relationship of eLH/CGR-eCG complexes in signal transduction.

## Figures and Tables

**Figure 1 ijms-22-10723-f001:**
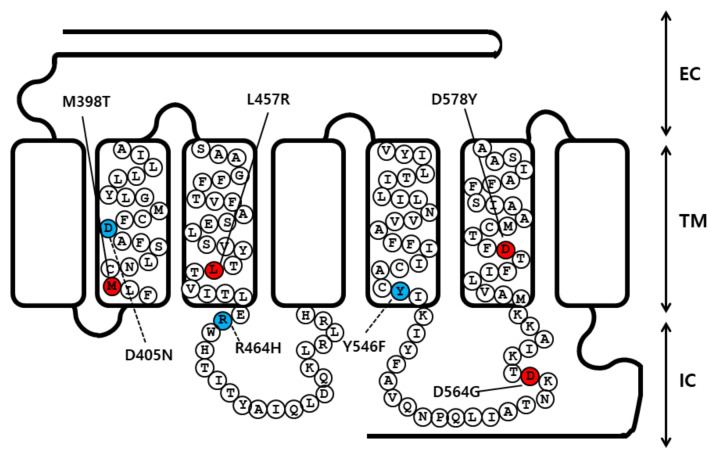
Schematic representation of the structure of eLH/CGR. The locations of the constitutive activating mutations (M398T, L457R, D564G, and D578Y) and the three inactivating mutations (D405N, R464H, and Y546F) are indicated. The red circle indicates the constitutively activating mutations, and the green circles indicate inactivating mutations. EC, extracellular domain; TM, transmembrane domain; IC, intracellular domain.

**Figure 2 ijms-22-10723-f002:**
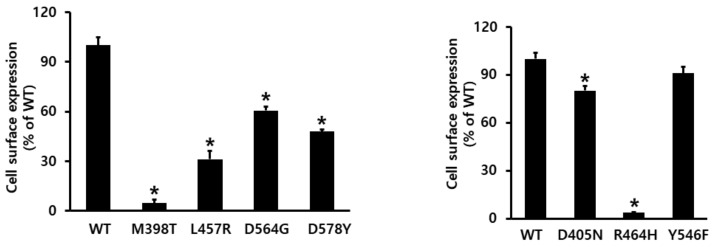
Cell-surface expression of eLH/CG receptors in transiently transfected HEK-293 cells. An enzyme-linked immunosorbent assay (ELISA) was used to determine the surface expression levels of the wild-type receptor and the indicated mutants of eLH/CGR. Data are presented as the means ± SEM of three independent experiments, and were normalized to the data for the wild-type. Cell-surface expression in the wild-type was considered as 100% (see Methods and Materials). * Statistically significant differences in cell-surface receptor expression (*p* < 0.05) compared to the expression of the wild-type receptor.

**Figure 3 ijms-22-10723-f003:**
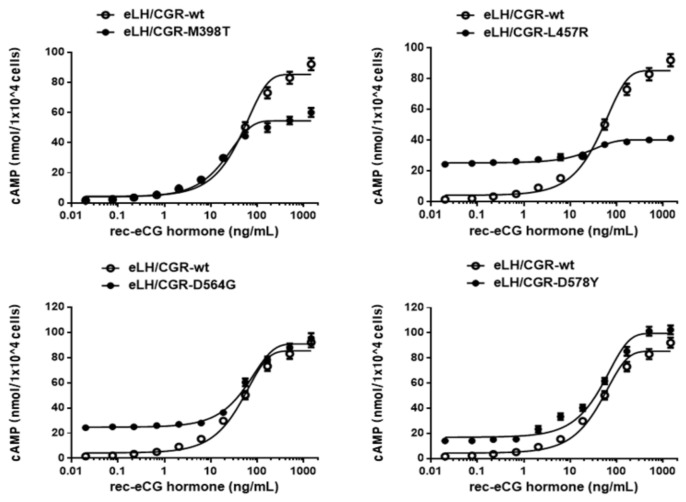
Total cAMP levels stimulated by recombinant eCG (rec-eCG) in CHO-K1 cells transfected with constitutively activating eLH/CGR mutants. CHO-K1 cells transiently transfected with wild-type eLH/CGR and mutants (M398T, L457R, D564G, and D578Y) were stimulated with rec-eCG in a medium containing 0.5 mM 3-isobutyl-1-methyl xanthine for 30 min. Levels of cAMP production were determined by homogeneous time-resolved fluorescence (HTRF). The cAMP accumulation was calculated as Delta F%. The cAMP concentration was recalculated and presented using the GraphPad Prism software. The results for the mock-transfected cells were subtracted from each data set. A representative data set was obtained from three independent experiments. The blank circles represent the corresponding curves for the wild-type receptor. Total cAMP was then measured in triplicate wells, as described in the Materials and Methods section. The figure depicts the results of the representative experiment performed with the indicated mutants. The lines are nonlinear least square fits of the experimental data.

**Figure 4 ijms-22-10723-f004:**
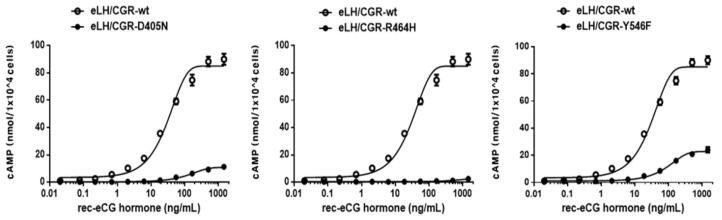
cAMP production stimulated by recombinant eCG (rec-eCG) treatment in CHO-K1 cells transfected with the inactivating eLH/CGR mutants. CHO-K1 cells transiently transfected with wild-type eLH/CGR and inactivating eLH/CGR mutants (D405N, R464H, and Y546F) were stimulated with rec-eCG for 30 min. Total cAMP accumulation was determined using homogeneous time-resolved fluorescence (HTRF). The empty circles denote wild-type eLH/CGR, and the black circles denote the mutants. The data were subtracted from the results of the mock-transfected cells. Representative data was obtained from three independent experiments.

**Figure 5 ijms-22-10723-f005:**
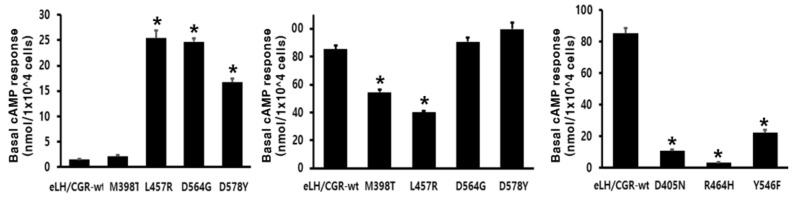
The results of basal cAMP responsiveness in activating mutants and Rmax level in activating/inactivating mutants. The basal and maximal AMP responses presented in Figure 3 and Figure 4 are displayed using a bar graph. * Statistically significant differences (*p* < 0.05) when compared with the wild-type receptor. A representative data set performed in triplicate out of two independent experiments. * Statistically significant differences in basal cAMP response and Rmax cAMP response (*p* < 0.05) compared to the expression of the wild-type receptor.

**Figure 6 ijms-22-10723-f006:**
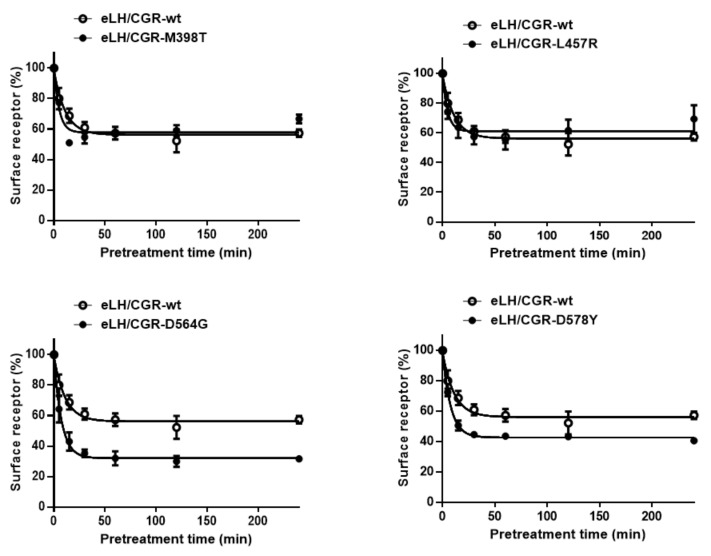
Time-dependent cell-surface loss in the wild-type eLH/CGR and activating eLH/CGR mutants. HEK-293 cells transiently expressing wild-type eLH/CGR or activating receptors (M398T, L457R, D564G, and D578Y) were incubated with 1000 ng/mL recombinant eCG (rec-eCG) for up to 240 min. Cell-surface expression in the non-pretreated groups was considered as 100% (see Materials and Methods for details). The loss of each receptor was determined using the GraphPad Prism software. The results are expressed as the means ± SE of three independent experiments. In this figure, the mean data are fitted to the one-phase exponential decay equation. The empty circles represent the corresponding curves for the wild-type receptor.

**Figure 7 ijms-22-10723-f007:**
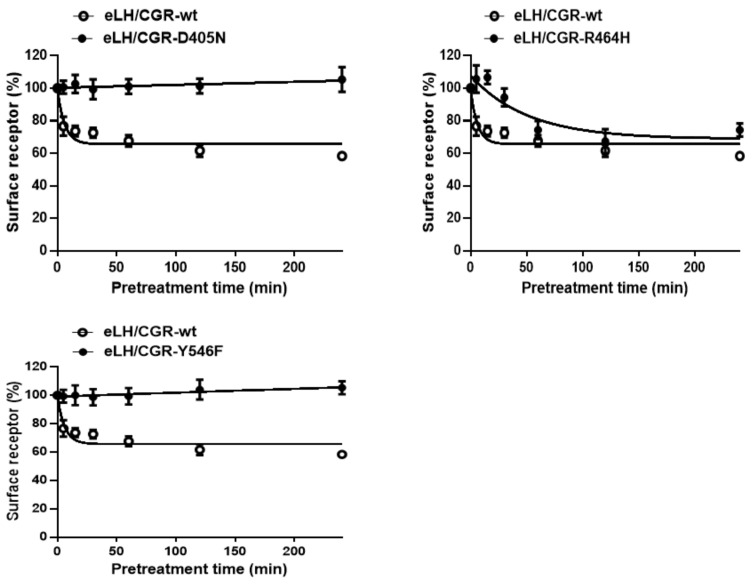
Time-dependent cell-surface loss in the wild-type eLH/CGR and inactivating eLH/CGR mutants. HEK-293 cells transiently expressing wild-type eLH/CGR or inactivating receptors (D405N, R464H, and Y548F) were incubated with 1000 ng/mL recombinant eCG (rec-eCG) for up to 240 min. Cell-surface expression in the non-pretreated groups was considered as 100% (see Materials and Methods for details). The loss of each receptor was determined using the GraphPad Prism software. The results are expressed as the means ± SE of three independent experiments. In this figure, the mean data are fitted to the one-phase exponential decay equation. The empty circles represent the corresponding curves for the wild-type receptor.

**Figure 8 ijms-22-10723-f008:**
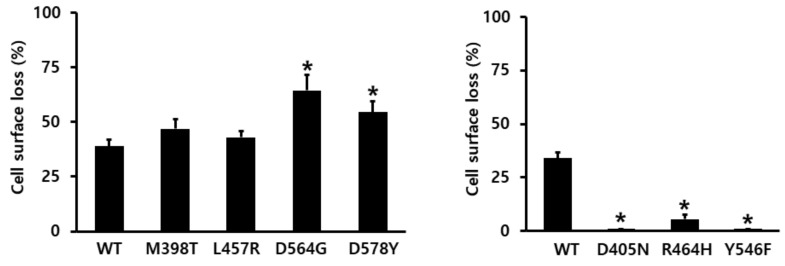
Cell-surface loss of wild-type eLH/CGR and activating/inactivating mutants. Each mutant plasmid was transiently transfected into the HEK-293 cells. The cells were incubated with or without 1000 ng/mL recombinant eCG (rec-eCG) for 30 min. Subsequently, the cell-surface expression of the receptors was determined. The results are expressed as percentages of the cell-surface loss of the receptor. Cell-surface losses of the wild-type and mutant eLH/CGRs were calculated by comparing the levels in the presence of rec-eCG to the levels in the absence of agonist treatment (considered as 0% cell-surface loss). The results are expressed as the means ± SE of three independent experiments. * Statistically significant differences in the cell-surface loss of the receptor (*p* < 0.05) compared to the cell-surface of the wild-type receptor.

**Table 1 ijms-22-10723-t001:** Bioactivity of eLH/CG receptors in cells expressing activating receptor mutants.

eLH/CG Receptors	cAMP Responses		
	Basal *^a^* (nmol/10 ^4^ Cells)	EC_50_(ng/mL)	Rmax *^b^* (nmol/10^4^ Cells)
eLH/CGR-wt	1.5 ± 0.2 (1.0-fold)	44.2 (38.1 to 52.7) *^c^*	85.3 ± 2.5 (100%)
eLH/CGR-M398T	2.1 ± 0.3 (1.4-fold)	19.8 (16.9 to 24.0)	54.5 ± 1.9 (64%)
eLH/CGR-L457R	25.4 ± 1.5 (16.9-fold)	24.4 (18.7 to 34.8)	40.2 ± 1.1 (47%)
eLH/CGR-D564G	24.7 ± 0.6 (16.4-fold)	58.3 (51.3 to 67.5)	90.8 ± 3.1 (106%)
eLH/CGR-D578Y	16.9 ± 0.8 (11.2-fold)	47.9 (40.5 to 58.8)	99.6 ± 4.5 (117%)

Values are the means ± SEM of triplicate experiments. The half maximal effective concentration (EC_50_) values were determined from the concentration-response curves from in vitro bioassays. The cAMP responses of the basal and EC_50_ in eLH/CGR wild-type were shown as onefold. Rmax response in eLH/CGR wild-type was also shown as 100%. *^a^* Basal cAMP level average without agonist treatment. *^b^* Rmax average cAMP level/10^4^ cells. *^c^* Geometric mean (95% confidence limit).

**Table 2 ijms-22-10723-t002:** Bioactivity of eLH/CG receptors in cells expressing inactivating receptor mutants.

eLH/CG Receptors	cAMP Responses		
	Basal *^a^* (nmol/10^4^ Cells)	EC_50_ (ng/mL)	Rmax *^b^* (nmol/10^4^ Cells)
eLH/CGR-wt	0.5 ± 0.1	30.3 (26.3 to 35.8) *^c^*	85.1 ± 3.5 (100%)
eLH/CGR-D405N	0.2 ± 0.1	134.9 (115.4 to 162.5)	10.8 ± 0.9 (13%)
eLH/CGR-R464H	1.9 ± 0.5	913.7 (662.4 to 1473)	3.4 ± 0.2 (4%)
eLH/CGR-Y546F	1.2 ± 0.3	95.9 (84.5 to 110.6)	22.3 ± 1.9 (26%)

The data are from three individual experiments. The half maximal effective concentration (EC_50_) values were determined from the concentration-response curves from in vitro bioassays. Rmax cAMP response of eLH/CGR wild-type was shown as onefold. *^a^* Basal cAMP level average without agonist treatment. *^b^* Rmax average cAMP level/10^4^ cells. *^c^* Geometric mean (95% confidence limit).

**Table 3 ijms-22-10723-t003:** Rates of cell-surface loss of receptors in transient transfected cell lines expressing the wild-type eLH/CGR and mutants thereof.

eLH/CGR Cell Lines	t_1/2_ (min)	Plateau (% of Control)
eLH/CGR-WT eLH/CGR-M398T eLH/CGR-L457R eLH/CGR-D564G eLH/CGR-D578Y	7.6 ± 0.6 (*n* = 8) 3.4 ± 0.2 (*n* = 4) 3.2 ± 0.1 (*n* = 6) 5.1 ± 0.3 (*n* = 5) 5.3 ± 0.4 (*n* = 4)	56.2 ± 3.1 57.7 ± 3.8 61.1 ± 3.2 32.1 ± 1.9 42.7 ± 2.9

Data were fitted to one phase exponential decay curves to obtain values of t*_1/2_* and plateau (i.e., maximum reduction). The data were collected from three individual experiments. Each value represents the mean ± SEM of the values from the indicated number of experiments.

**Table 4 ijms-22-10723-t004:** Rates of cell-surface loss of receptors in transient transfected cell lines expressing the wild-type eLH/CGR and mutants thereof.

eLH/CGR Cell Lines	t_1/2_ (min)	Plateau (% of Control)
eLH/CGR-WT eLH/CGR-D405N eLH/CGR-R464H eLH/CGR-Y546F	5.3 ± 0.3 (*n* = 6) -^a^ (*n* = 3) 35.6 ± 2.1 (*n* = 4) - (*n* = 3)	65.6 ± 2.7 - 68.7 ± 4.5 -

Data were fitted to one phase exponential decay curves to obtain values of t*_1/2_* and plateau (i.e., maximum reduction). The data were collected from three individual experiments. Each value represents the mean ± SEM of values from the indicated number of experiments. ^a^ nondetectable.

**Table 5 ijms-22-10723-t005:** List of primers used to construct eLH/CGR mutants.

	Primer Name	Primer Sequence
1	eLH/CGR-wt forward	5′-ATGAATTCATGGGGAGAAGGTCACTAGCACTAC-3′ EcoRI site
2	eLH/CGR-wt reverse	5′-CCTCGAGTTAACACTCTGTATAGCAAGTCTT-3′ XhoI site
3	M398T forward	5′-CTAACAGTGCCCCGTTTTCTCACGTGCAATC-3′
4	M398T reverse	5′-GATTGTCACGGGGCAAAAGAGTGCACGTTAG-3′
5	L457R forward	5′-CTGCTACACCCGCACAGTCATCACACTAG-3′
6	L457R reverse	5′-GACAGATGTGGGCGTGTCAGTAGTGTGATC-3′
7	D564G forward	5′-CTTAGCAATCTTTGTGCCTTTGTTGGTAGC-3′
8	D564G reverse	5′-GCTACCAACAAAGGCACAAAGATTGCTAAG-3′
9 10 11	D578Y forward D578Y reverse D405N forward	5′-CCTCATCTTCACCTATTTCACCTGCATGGCACC-3′ 5′-CCATGCAGGTGAAATAGGTGAAGATGAGGACTGC-3′ 5′-CTCTCTTTTGCAAACTTTTGCATGGGGCTCTATC-3′
12	D405N reverse	5′-GCCCCATGCAAAAGTTTGCAAAAGAGAGATTGCA-3′
13	R464H forward	5′-CACACTAGAACACTGGCACACCATCACCTATG-3′
14	R464H reverse	5′-GATGGTGTGCCAGTGTTCTAGTGTGATGACTGTG-3′
15	Y546F forward	5′-GTGCTTGCTTCATTAAAATTTATTTTGCAG-3′
16	Y546F reverse	5′-TTAATGAAGCAAGCACAAATGATGAAGAAGGC-3′

Underlined nucleotides are the site of mutagenesis. The mutant and wild-type eLH/CGR cDNAs were subcloned into the eukaryotic expression vector pcDNA3 and pCORON1000 SP VSV-G for transfection. The plasmids were purified, and the presence of the correct insert was confirmed by restriction enzymes. Finally, we constructed eight receptor genes: wild-type eLH/CGR, M398T, L457R, D564G, D578Y, D405N, R464H, and Y546F.

## Data Availability

Not applicable.

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
