# Peer review of "Constitutively Activating Mutants of Equine LH/CGR Constitutively Induce Signal Transduction and Inactivating Mutations Impair Biological Activity and Cell-Surface Receptor Loss In Vitro"

_ijms, 2021, doi:10.3390/ijms221910723_

Round 1

Reviewer 1 Report

The authors describe the effects of introducing four constitutively activating and three constitutively inactivating mutations in the eLH//CGR on surface expression, receptor activation as indicated by cAMP accumulation, and receptor internalization. It looks like one activating and one inactivating mutant exhibited high surface expression, minimal cAMP increase, and influenced receptor internalization consistent with their expected activities. The results with two inactivating mutations are consistent with the notion that only ligand-activated receptors are internalized, as none was observed over 4 hr. The increased rates of constitutively active receptor internalization appear consistent with prior activation of receptor internalization machinery prior to ligand addition. The experiments were for the most part well described. However, the surface receptor quantification method was obscure. It took several reads to determine that an epitope tag had been added to the eLHCGRs, as the modifications were never mentioned in the results.

The manuscript is very well written with a few minor suggestions made below:

Specific comments.

Line 54:  possesses should replace “occupies”.

Line 185:  delete space in 117%

Line 294:  folds should be singular

Line 527:  either “completely” should be changed to “complete”, or the word that completely is modifying is missing.

Author Response

Reviewer 1

Specific comments

Line 54: possesses should replace “occupies”

®We changed “occupies” to ‘possesses” in the Line 52.

Line 185: delete space in 1 17%

®We deleted “space” in the 1 17% in the Line 189. ‘

Line 294: folds should be singular

®We changed “folds” to “fold” in the Line 304.

Line 527: either “completely” should be changed to “complete”, or the word that completely is modifying is missing.

®We changed “completely” to “complete” in the Line 544.

Reviewer 2 Report

The review on the original paper

Munkhzaya Byambaragchaa1 and coauthors ”Constitutively Activating Mutants of Equine LH/CGR Consti-2 tutively Induce Signal Transduction and Inactivating Muta-3 tions Impair Biological Activity and Cell-Surface Receptor Loss 4 In Vitro”

The article is devoted to the topical problem of molecular endocrinology and reproductology - the study of the structural and functional organization of LH/hCG receptors and the study of molecular mechanisms of activation and internalization of mutant forms of these receptors. In the present study, the authors examined in details seven mutant forms of the equine LH/hCG receptor, four of which had activating mutations and three more inactivating mutations. At the same time, activating mutations increased the basal level of cAMP in cells with expressed mutant LH/hCG receptors, modulated the cAMP response to gonadotropins, and accelerated the process of LH/hCG receptor internalization and a decrease in the number of receptor molecules on the surface of the target cell. Inactivating mutations impaired signal transduction through the equine LH/hCG receptor-Gs protein-adenylate cyclase system, but had a little effect on the surface density of the mutant receptors. The key role of gonadotropin-stimulated cAMP signaling in the regulation of the processes of internalization and recyclization of receptors has been shown. The methods used are fully adequate to the tasks, the conclusions correspond to the obtained results. It should be emphasized that the presented article is an important part of a comprehensive study by the authors, the focus of which is on mutant forms of LH/hCG receptors in various species of vertebrates.

However, there are a number of questions and comments on the article.

  1. The authors point out that the M398T mutation in the eLH/hCG receptor increases the basal adenylate cyclase activity, but in fact no significant changes in the basal cAMP level were found (2.1 in the M398T mutant versus 1.5 nM/10000 cells in cells with a wild-type receptor). Moreover, the basal cAMP level in Table 1 (experiments with activating mutations) is 1.5 nM/10000 cells, while in Table 2 (inactivating mutations) it is 0.5 nM/10000 cells. At the same time, the authors did not indicate an increase in the basal activity in the mutant with the inactivating mutation R464H (1.9 nM/10000 cells), although the differences with the cells expressing the wild type receptor in this case are more significant. It is also necessary to explain the significant difference in EC50 values ​​for cells with wild-type receptors in Tables 1 and 2. For cells expressing wild-type receptors, there are also differences in values ​​of t1/2 and plateau (maximum reduction) (Tables 3 and 4). Although such differences are perfectly acceptable, they need to be discussed in the text.
  2. The authors write (lines 331-333): “According to the present results, the rates of cell-surface loss of receptor in all activating mutants (3.2∼5.3 min) were faster than that observed in cells expressing the wild-type receptor (7.6 min), clearly indicating important correlations between cAMP response and cell-surface receptor loss ”. But, as noted in point 1, there is no significant increase in the level of cAMP in the M398T mutant, and, therefore, cAMP-independent mechanisms of receptor loss should be indicated and discussed.
  3. Basal activity values ​​are given in units of nM/10000 cells. How true is this? The dimension "nM" indicates the concentration of the substance, and the concentration cannot be attributed to the number of cells. The ratio of the amount of cAMP (for example, nmols) to the number of cells (or mg of protein, mg of the tissue, etc.) should be indicated. It should be noted that the manufacturer's instructions for the kit for determining the cAMP level do not clearly describe the units of measurement, but the dimension nM/10000 cells, from my point of view, is meaningless. Please explain the calculations in more detail (it is better to give specific examples of calculations).
  4. The authors point out that the activation of the LH/hCG receptor leads to the activation of cAMP signaling (and this is indeed the main pathway induced by LH and even more so by hCG), but beta-arrestin pathways and the LH/hCG receptor-Gq/11 protein-phospholipase Cbeta-calcium signaling pathway are also activated. In this case, the beta-arrestin pathway plays a key role in down-regulation of the eLH/hCG receptor and its internalization. Why these pathways are completely ignored in the discussion, because it is the participation of these pathways that can explain the significant differences identified by the authors in the specific activity of the studied mutant forms of the eLH/hCG receptor, as well as in their effect on the loss of receptors on the cell surface (for example, eLH / CGR-M398T and eLH/CG -L457R, or eLH/CGR-D405N or -R464H and eLH/CGR-Y546F) (see also point 2). From this point of view, the discussion should place greater emphasis on the revealed differences in the biological activity of various mutants.
  5. The authors investigate mutations of the eLH/hCG receptor of different localization, and rightly point out their possible role in the binding of ligands and heterotrimeric G-proteins. But at the same time, the substituted amino acids located in the transmembrane regions can participate in the formation of the transmembrane allosteric site of the LH/hCG receptor, with which low molecular weight allosteric regulators and modulators bind. At the same time, amino acids of different localization in intracellular loops and in ICL/transmembrane interfaces are involved in interaction with different types of G-proteins. It is desirable to discuss this.
  6. It is necessary to add information on the presentation of results to tables and figures (n, SEM, etc.)
  7. Line 185 - “1 17%” replaced by “117%”

Author Response

Reviewer 2

  1. The authors point out that the M398T mutation in the eLH/hCG receptor increases the basal adenylate cyclase activity, but in fact no significant changes in the basal cAMP level were found (2.1 in the M398T mutant versus 1.5 nM/10000 cells in cells with a wild-type receptor). Moreover, the basal cAMP level in Table 1 (experiments with activating mutations) is 1.5 nM/10000 cells, while in Table 2 (inactivating mutations) it is 0.5 nM/10000 cells. At the same time, the authors did not indicate an increase in the basal activity in the mutant with the inactivating mutation R464H (1.9 nM/10000 cells), although the differences with the cells expressing the wild type receptor in this case are more significant. It is also necessary to explain the significant difference in EC50 values ​​for cells with wild-type receptors in Tables 1 and 2. For cells expressing wild-type receptors, there are also differences in values ​​of t1/2 and plateau (maximum reduction) (Tables 3 and 4). Although such differences are perfectly acceptable, they need to be discussed in the text.

®We deleted that the M398T mutant exhibited 1.4-fold in basal cAMP response in the Line 26 of Abstract.

®We deleted that the M398T mutant exhibited 1.4-fold in basal cAMP response in the Line 139-140 of Result.

®We inserted “however, M398T mutant was not induced.” In the Line 141.

®We also inserted “The difference in basal cAMP response and EC50 values for the wild-type receptors in Table 1 and 2 was attributed to the fact that the experiments were performed separately.” In the Line 193-194.

®We also inserted “For cells expressing wild-type receptors, the values of t1/2 and maximum reduction presented in Table 3 and 4 were different  because the experiments were conducted independently.” In the Line 279-281.

  1. The authors write (lines 331-333): “According to the present results, the rates of cell-surface loss of receptor in all activating mutants (3.2∼5.3 min) were faster than that observed in cells expressing the wild-type receptor (7.6 min), clearly indicating important correlations between cAMP response and cell-surface receptor loss”. But, as noted in point 1, there is no significant increase in the level of cAMP in the M398T mutant, and, therefore, cAMP-independent mechanisms of receptor loss should be indicated and discussed.

®We inserted that “Specifically, M398T mutant did not exhibit any increase in the basal cAMP response; however, t1/2 value for this mutant indicated 2.2-fold faster loss of cell-surface receptor compared to those of wild-type receptor. Thus, these differences between basal cAMP response and cell-surface loss of receptors need to be clearly demonstrated.” In the Line 345-349.

  1. Basal activity values ​​are given in units of nM/10000 cells. How true is this? The dimension "nM" indicates the concentration of the substance, and the concentration cannot be attributed to the number of cells. The ratio of the amount of cAMP (for example, nmols) to the number of cells (or mg of protein, mg of the tissue, etc.) should be indicated. It should be noted that the manufacturer's instructions for the kit for determining the cAMP level do not clearly describe the units of measurement, but the dimension nM/10000 cells, from my point of view, is meaningless. Please explain the calculations in more detail (it is better to give specific examples of calculations).

®We changed “cAMP nM (1X10^4 cells)” to “ cAMP (nmol/1x10^4 cells)” in the all Figure and Table.

  1. The authors point out that the activation of the LH/hCG receptor leads to the activation of cAMP signaling (and this is indeed the main pathway induced by LH and even more so by hCG), but beta-arrestin pathways and the LH/hCG receptor-Gq/11 protein-phospholipase Cbeta-calcium signaling pathway are also activated. In this case, the beta-arrestin pathway plays a key role in down-regulation of the eLH/hCG receptor and its internalization. Why these pathways are completely ignored in the discussion, because it is the participation of these pathways that can explain the significant differences identified by the authors in the specific activity of the studied mutant forms of the eLH/hCG receptor, as well as in their effect on the loss of receptors on the cell surface (for example, eLH / CGR-M398T and eLH/CG -L457R, or eLH/CGR-D405N or -R464H and eLH/CGR-Y546F) (see also point 2). From this point of view, the discussion should place greater emphasis on the revealed differences in the biological activity of various mutants.

® We inserted “β-arrestin, ubiquitously expressed intracellular regulators of GPCR trafficking, pathway plays a key role in the down-regulation of the GPCR and its internalization. G protein and β-arrestin signaling mediate distinct physiological effects [42,43]. However, β-arrestin activation by eLH/CGR has not been studied.” In the Line 390-394.

  1. The authors investigate mutations of the eLH/hCG receptor of different localization, and rightly point out their possible role in the binding of ligands and heterotrimeric G-proteins. But at the same time, the substituted amino acids located in the transmembrane regions can participate in the formation of the transmembrane allosteric site of the LH/hCG receptor, with which low molecular weight allosteric regulators and modulators bind. At the same time, amino acids of different localization in intracellular loops and in ICL/transmembrane interfaces are involved in interaction with different types of G-proteins. It is desirable to discuss this.

® Recently, biased allosteric modulators and regulators of other GPCR have been reported [40,41,44]. In the present study, the mutants located in the transmembrane domain and between intracellular loops and transmembrane domain demonstrated that eLH/CGRs may be involved in interaction with different types of G-proteins.” In the Line of 395-399.

  1. It is necessary to add information on the presentation of results to tables and figures (n, SEM, etc.)

®We inserted “Total cAMP was then measured in triplicate wells, as described under “Materials and Methods.” The figure depicts the results of the representative experiment performed with the indicated mutants. The lines are nonlinear least square fits of the experimental data.” In the Line 166-168.

®The data were corrected from three individual experiments. Each value represents the mean ± SEM of the values from the indicated number of experiments.” In the Line 256-286 and Line 290-291 of the Table 3 and 4.

  1. Line 185 - “1 17%” replaced by “117%”

® We changed “1 17%” to “117%”. In the Line 189.
